# Assessing the Impact of Air Pollution on Inbound Tourism along the Yangtze River across Space and Time

**DOI:** 10.3390/ijerph191710944

**Published:** 2022-09-02

**Authors:** Xiang Zhang, Chenjiao Ma, Xingming Li, Lina Xiong, Silin Nie

**Affiliations:** 1Wuhan Branch of China Institute of Tourism, Central China Normal University, Wuhan 430079, China; 2Key Laboratory for Geographical Process Analysis & Simulation of Hubei Province, Central China Normal University, Wuhan 430079, China; 3College of Urban and Environmental Science, Central China Normal University, Wuhan 430079, China; 4Department of Human Dimensions of Natural Resources, Colorado State University, Fort Collins, CO 80523, USA

**Keywords:** air pollution, inbound tourism, fixed effect model, Yangtze River Economic Belt, environmental impact

## Abstract

The prevalent air pollution along the Yangtze River Economic Belt (YREB) possesses a significant threat to the natural environment, which further affects nearby tourism destination development. The paper seeks to assess the impact of air pollution on tourism in this region through a 2002–2012 panel data of 31 prefecture-level cities, along with geographic information system (GIS) and cluster analyses. The results reveal that air pollution is negatively associated with the number of inbound tourists along the YREB. In general, when air pollution intensifies by 1%, the number of inbound tourists decreases by 1.171%. This impact is more evident when air pollution is more severe, in the long term, and in areas that are larger, more central, and with more tourism resources. The paper contributes to the literature by addressing common limitations in previous studies and providing a more comprehensive evaluation of air pollution’s impact on inbound tourism in the YREB. Practical implications regarding public policies and development directions based on air pollution periods, regions, and tourism resource allocations are provided.

## 1. Introduction

Air pollution has become a serious concern along with rapid industrialization, urbanization, and remarkable economic achievement in China [1]. In November 2013, the Chinese Academy of Social Sciences and China Meteorological Administration jointly released a report titled “The Green Paper on Climate Change: Report on Tackling Climate Change”. This report suggested that smog has become increasingly frequent in the past 50 years [2]. The State of China’s Ecological Environment also suggested that 239 of 338 prefecture-level (70.7%) cities did not meet the air quality standards in 2017. The tourism industry is considered one of the most sensitive industries that are susceptible to damage to the natural environment. Considering the wide range of the negative impact of air pollution (e.g., increasing respiratory illness risk, endangerment to wildlife, acid rain, industry and school closures), it is highly relevant and meaningful to examine how air pollution may affect the tourism industry [3].

Specifically, air pollution can directly affect the spatial and temporal distribution of tourism resources, tourist behaviors, operating costs, etc. It can also affect tourists’ aesthetic experiences and perceived images of destinations, which can further hinder the sustainable development of tourism destinations [4]. It also indirectly affects the tourism industry through its impact on the broader social and economic environment [5]. This paper focuses on the impact of air pollution on inbound tourism in China, which is an indispensable part of the total tourism market. In this paper, inbound tourism refers to international tourists traveling to Chinese destinations, such as Japanese tourists visiting the Three Gorges Dam in Hubei, China. Overall, scholars have suggested that air pollution negatively affects inbound tourism in China. For instance, the China Tourism Academy identified air pollution as the main reason for the declining inbound tourism in China in their 2014 annual tourism report. International media outlets have also reported noxious air quality in China and linked it to the decreasing destination appeal of China as an international destination [6].

Although it is intuitive to recognize the negative relationship between air pollution and inbound tourism, there is a lack of an in-depth understanding of this relationship due to different measures and research methods used in previous studies. For example, reports from China’s National Bureau of Statistics regarding air pollution (the total emissions of SO_2_, NOx, and smoke dust) and inbound tourism (number of inbound tourists) from 2011 to 2017 showed inconclusive patterns. During this period, air pollution showed a consistent downward trend (except for a minor increase in 2014) suggesting an improving air quality. However, the number of inbound tourists did not follow this trend consistently. Rather, it showed a decreasing trend from 2011 to 2014, and then an increasing trend afterwards. Some scholars have studied the impact of air pollution on inbound tourism using data from China’s National Bureau of Statistics but with inconclusive results. For example, Zhan and Yin (2007) [3] found that there was a decreasing trend in air pollution levels in China from 2005 to 2012, but this change did not have a significant impact on tourism. In contrast, Yan and Zhang (2016) [2] found that from 2004 to 2011, the level of air pollution decreased while the number of inbound tourists increased, and air pollution showed a significant impact on inbound tourism. Although there could be other factors that have contributed to the increase in inbound tourism since 2014, it is also likely that there might be time lags between air pollution and its impact on inbound tourism. For instance, the lag of effects could be influenced by seasonality. Air quality differs significantly from season to season. In winter, air quality might be the lowest because of increased energy use (e.g., coal burning). In addition, tourism is also susceptible to seasonality in that late spring, summer, and early fall tend to be the busiest time for tourism. In addition, there is also a lack of consideration of factors that are highly relevant to tourism development, such as location and the concentration of tourism resources.

To fill this gap, the authors of this paper conducted an in-depth examination of the impact of air pollution on inbound tourism in the area of the Yangtze River Economic Belt (YREB) in China. It is a national strategic economic zone that promotes the coordinated development of the regional economy. This region covers nine provinces and two municipalities in China, with a total of 126 cities. They include the Shanghai municipality, Jiangsu province, Zhejiang province, Anhui province, Jiangxi province, Hubei province, Hunan province, the Chongqing municipality, Sichuan province, Guizhou province, and Yunnan province. The YREB experienced significant growth in tourism development. According to provincial statistical reports, the YREB received 41,558,100 inbound tourists, accounting for 35.87% of China’s inbound tourists in 2017. From 2013 to 2017, the annual growth rate of inbound tourists in the YREB was 5.47 percent, compared to 4.97 percent nationwide [7]. This region also experienced significant economic growth from 2010 to 2017, with a 113.9 percent increase in GDP (from about USD 2.65 trillion to USD 5.66 trillion). However, much of the economic growth was driven by pollution-heavy manufacturing industries, such as thermal power generation, petrochemicals, and steel. As a result, air pollution became a serious concern. For example, fewer than one-third (33 cities) of the 126 cities in the YREB met the standard level of average concentrations of air pollutants (SO_2_, NO_2_, PM_10_, PM_2.5_, CO, and O_3_) in China in 2016 [8]. Although researchers have suggested an interplay effect between air pollution and tourism [9], considering the emphasis on economic growth and the industry structure in the YREB, the YREB was chosen as a representative case to examine the complicated impact of air pollution on inbound tourism. Using a fixed effect model, we measured the impact of air pollution on inbound tourism in the YREB as a whole and in different regions, as well as during different times. The findings provide valuable insights and guidance for local governments and tourism operations for a more sustainable destination future as well as for similar destinations worldwide.

## 2. Literature Review

The relationship between air pollution and inbound tourism has been discussed extensively in the literature [9,10,11,12], there is even a review of the impact of air quality on tourism from the perspective of tourism demand [13]. Nevertheless, the results are inconclusive. The discrepancy can be attributed to three factors, including the timing of the research, regional differences, and the pollutants considered. Although it is generally argued that air pollution negatively affects inbound tourism, some studies have also identified that inbound tourists are not budged by air pollution. For example, Law and Cheung (2007) [14] showed that international tourists did not consider air pollution as an issue when choosing to travel to Hong Kong. However, their neutral attitude changed after they traveled to Hong Kong. This discrepancy might be explained by the timing of studies, as tourists may experience more negative effects of air pollution when they stay for a long time at the destination. From a more longitudinal perspective, Geng et al. (2021) [15] showed that the relationship between air pollution and inbound tourism is relatively stable in most areas of China with small fluctuations. Liu et al. (2018) [4] examined three time periods including 2001–2005, 2006–2010, and 2011–2015, and suggested that air pollution had an overall negative impact on the development of inbound tourism in China, and the marginal effect increased over time. That is, tourists were more likely to perceive the negative effects of air pollution when they spent more time at the destination, as well as when air pollution worsened during their trips. Although Liu et al. (2018) [4] considered different time periods, these periods were decided based on the Chinese national economy planning timeline, which could be different from real economic development/air pollution periods. Therefore, it is likely that the impact of air pollution on inbound tourism may differ depending on a short- or long-term perspective.

In addition, in consideration of air pollution diffusion, regional heterogeneity is also a factor that may explain different impacts of air pollution on inbound tourism in different regions [4,12,16]. Because of the vast areas covered in the YREB, together with the Yangtze River appeal, the YREB provides a variety of tourism products such as river cruises, city attractions (e.g., museums and parks), historical sites (e.g., the Yellow Crane Tower), and geographic wonders (e.g., the Three Gorges National Geopark). It is no surprise that the YREB experienced significant growth in tourism development. According to provincial statistical reports, the YREB received 41,558,100 inbound tourists in 2017. Xie et al. (2017) [16] studied the impact of air pollution on inbound tourism using China’s provincial panel data from 2005 to 2013. The results show distinct impacts of smog on inbound tourism in different areas. Among them, the eastern region experienced the greatest negative impact, followed by the central region and the western region (which is mostly unaffected). The northern region is more negatively affected by PM_10_, and the southern region is more negatively affected by SO_2_ and smoke dust. Along the same line, Liu et al. (2018) [4] divided mainland China into higher, high, low, and lower air pollution areas according to the degree of air pollution. They found that all levels of air pollution had a significant negative impact on inbound tourism. The negative impact was more pronounced in areas with higher pollution levels. In the global context, studies on the impact of air pollution on tourism also received inconclusive results. Zhang et al. (2019) [17] employed the autoregressive distributed lag (ARDL) statistical method to study the relationship between tourism and environmental degradation and suggested that air pollution had a negative impact on inbound tourism in Thailand. Faiza et al. (2014) [10] identified the distinct impact of air pollution on tourism in different regions. For example, in sub-Saharan Africa, air pollution and international tourism expenditure were negatively correlated. However, in the Middle East and North Africa, air pollution and international tourism expenditure were positively correlated. Although this regional heterogeneity factor has been shown to affect the relationship between air pollution and inbound tourism, these studies have often been conducted across countries and prominent regions within a country. Specific studies on the growing YREB area are lacking. In addition, categorizing areas based on the levels of tourism resource concentration may offer further insights into how air pollution’s impact on inbound tourism may differ.

Lastly, given the different pollutants included in previous studies, the findings are often specific to the selected pollutants. Ye et al. (2021) [12] investigated three major pollutants, including SO_2_, NO_2_, and smoke dust in the air. They found that although all of them had a significant negative impact on inbound tourism revenue, the impact levels were different. Specifically, SO_2_ had the greatest impact, followed by NO_2_ and smoke dust. Zhan and Yin (2017) [3] found that PM_10_ did not have a significant negative impact on inbound tourism by examining the mass concentration of PM_10_ in the air. Fang et al. (2020) [18] suggested that there is a certain “inverted U-shaped” relationship between tourism economic development and SO_2_ emissions, and a weak “U-shaped” relationship between tourism economic development and smoke dust emissions. In contrast, Sun (2020) [19] found that the negative impact of air pollutants on tourism can be ranked as PM_2.5_, PM_10_, SO_2_, CO, and NO_2_ in descending order. Therefore, a rigorous examination of air pollution’s impact on inbound tourism further needs to consider the inclusion of the main pollutants (e.g., the use of an air pollution index) to generate a holistic view.

In summary, although many studies have examined the impact of air pollution on inbound tourism, they have often generated inconsistent results because of their research methods regarding the time periods, regions, and pollutants considered. The YREB is an ideal study site to observe these potential differences because of its vastly different cities with respect to tourism resources, economic growth, and air pollution. Based on the identified limitations in prior research, we applied three corresponding conditions. First, we identified the research periods based on the natural diffusion of air pollution. Second, we considered the regional heterogeneity based on tourism resource distribution. Third, we adopted a comprehensive evaluation index of air pollutants to analyze the impact of air pollution on inbound tourism. The following sections describe how we evaluated the relationship between air pollution and inbound tourism in the YREB with a thorough consideration of evaluation periods, tourism resource distribution, and a comprehensive air pollutant index.

## 3. Research Methods

### 3.1. Research Area—The Yangtze River Economic Belt (YREB)

The YREB is the area along the Yangtze River and is rich in natural, geographical, and cultural resources. It includes 11 provincial and 126 prefecture-level cities, connecting the less developed west and the more developed east coastal areas (Figure 1). The YREB area covers about 21.28% of the land area in China. About 42.87% of the Chinese population live here, contributing to about 42.23% of the Chinese economy.

In addition, the population density in the YREB is higher than that of China overall. From 2002 to 2012, the population density of the YREB was about 1.7 times the national level, which may drive stronger economic growth as well as more tourists in this region. Other control factors, such as the high-speed railway system, which grew significantly in recent years, should be considered in future research as it facilitates more long-distance travel. Population density, which varies among cities included in this study (as well as from the national level), may serve as a confounding factor that is encouraged to be controlled in future models. Thus, the YREB plays an extremely important role in China’s national development strategies [20]. However, there are significant huge economic development gaps among the provinces and cities, where the east region is generally more prosperous than the west.

Along with the significant economic growth, the YREB has experienced major air pollution issues that far exceeded the capacity of the environment. In 2012, the emissions of SO_2_, NO_x_, and smoke dust in the YREB (population: 583 million) were 7.34 million tons, 7.59 million tons, and 3.46 million tons, respectively. In comparison, about 4.74 million tons of SO_2_ emissions were produced in 2012 in the entire US, with a population of 310 million. In the same year, the SO_2_ emissions were 0.94 million tons for Japan (population: 127.48 million), 0.43 million tons for Germany (population: 81.6 million), and 15.628 million tons for the Organization for Economic Cooperation and Development (OECD). The per capita SO_2_ emissions in the YREB were lower than that of the US (83.7 percent), but higher than those of Japan (1.7 times) and Germany (2.4 times), and similar to the level of the OECD. In addition, the intensity of pollutants per unit area in the YREB was 1.6 times, 1.5 times, and 1.3 times the Chinese national average, respectively. Some cities in the center and east regions of the YREB have experienced over 100 days of haze per year and some cities experienced over 200 days of haze, suggesting significantly low air quality [21]. To combat the air quality issue, China began to implement a more aggressive air quality standard in 2013. Compared to the old standard, the new standard updated the pollutant list and pollutant limits, added the average concentration limits of PM_2.5_ and O_3_, tightened the concentration limits of PM_10_, NO_2_, and other pollutants, and updated the analytical method standards of SO_2_, NO_2_, O_3_ particles, etc. It was identified as a main driver of the improved air quality from 2013 to 2017 [22]. Before 2013, Chinese governments at all levels used the air pollution index (API) to assess air pollution. Compared to the AQI, the API integrates several air pollutants, such as SO_2_, NO_x_, and total suspended particulate matter, into a single conceptual index value to account for air pollution levels. A detailed estimation method is shown in Fan (1998) [23]. Although the API and AQI are comparable indices, the API statistics were readily and consistently available for the study period from China’s National Bureau of Statistics. Therefore, we focused on the results based on the API. In 2013, Chinese governments started using the air quality index (AQI) suggested in the new ambient air quality standard for evaluation. These two sets of evaluation tools (API and AQI) are distinct due to different evaluation criteria and pollutant indicators adopted. The paper focuses on the 2002–2012 period and adopts the API to evaluate the degree of air pollution.

Although there are 126 prefecture-level cities in the YREB, only 31 prefecture-level cities released air pollution data with unified air quality standards (Figure 2). Given that at least two cities in each province are included and these cities are dispersed across the YREB, the evaluation of the impact of air pollution on inbound tourism in these selected cities should be representative of the YREB. In addition, the selected cities also play a significant role in regional economic and tourism growth. For instance, the total GDP of the 31 cities accounted for 43.66% of the YREB in 2012, and the number of inbound tourists accounted for 68.52%.

Figure 2 provides an overview of the API levels of the 31 cities from 2002 to 2012. The 31 cities are divided into three parts. The first part includes 10 cities in the upper Yangtze River, the second part includes 9 cities in the middle Yangtze River, and the third part includes 12 cities in the lower Yangtze River. In general, the API levels decreased during this period. The discrepancy among the API level changes is likely due to different local government guidance. Cities with improved air quality often adopted action plans to combat air pollution (e.g., Chongqing), encourage commercial businesses and service industries (e.g., Suzhou), and develop eco-tourism (e.g., Zhangjiajie). Cities that experienced worsened air quality (e.g., Jingzhou) tended to prioritize high-pollution manufacturing industries (e.g., chemical, textile) in economic development.

Figure 3 further presents the spatial distribution of air pollution levels and population density. In general, the air pollution level is higher in the east and lower in the west (Figure 3). Among them, areas with low API values (i.e., better air quality) are mainly in the upper Yangtze River. This is likely due to a low level of industrialization. Areas with high API values are often in major cities, such as Chengdu, Wuhan, and Changsha, which are capital cities as well as population centers. These areas also experienced a higher level of urbanization and industrialization, thus producing more air pollution.

### 3.2. Research Methods

#### 3.2.1. Variable Selection

Previous studies have widely used the number of inbound tourists and tourism income to measure the development of inbound tourism. Following this approach, the authors of this paper also selected the number of inbound tourists and tourism income as the indicators to measure the development level of inbound tourism, which is modeled as the outcome variable. The core explanatory variable is the degree of air pollution. As mentioned before, the air pollution index (API) was used to measure the degree of air pollution.

Control variables, including the levels of economic development, tourism resource endowment, tourism reception capacity, and transportation, were adopted to provide more accurate results. These control variables were selected because they may affect inbound tourism development based on previous literature [24,25,26,27,28].

Specifically, economic development promotes the development of inbound tourism by enhancing the international influence of cities, increasing funds for tourism facility construction, and promoting import and export trades [24,25]. In this paper, the economic development level is measured by the per capita GDP of each selected city in 2002. Tourism resources are the foundation of tourism destination development and a key driver of tourists. Tourists in inbound tourism often seek a higher level of tourism experiences, and therefore, areas with more prominent attractions are likely to welcome more international tourists [26,27]. In this paper, the level of tourism resource endowment was measured by the number of 4A and 5A attractions in each city (5A is the highest level of attraction quality designated by the Ministry of Culture and Tourism in China). Following previous research [26], this paper gives 2.5 points to 4A attractions and 5 points to 5A attractions. The sum score was used as the proxy variable of tourism resource endowment. Subsequently, the sample cities were divided into three categories for heterogeneity analysis: Class A cities (cities with high tourism resource endowment) have a tourism resource endowment score of more than 40; class B cities (cities with medium tourism resource endowment) have a tourism resource endowment score of 20–40; class C cities (cities with low tourism resource endowment) have a tourism resource endowment score of less than 20.

Tourism reception capacity reflects the level of destinations’ abilities to host visitors. One key indicator of this capacity is the number of hotels, especially for inbound tourists, as they often need to spend multiple days at destinations and require lodging services of a higher standard [1]. Therefore, the number of star-rated hotels in each region was used to reflect tourism capacity. A higher number suggests a higher level of tourism reception capacity. Last but not the least, transportation plays a critical role in guiding tourist flows and affects the number of tourists in each region. Better transportation (e.g., number of travel options, quality, and convenience of routes) facilitates tourism development. Given that inbound tourists primarily use highways to travel in prefecture-level cities of the YREB during the study period [1], the highway network density was adopted to reflect the transportation conditions. It was estimated by dividing the total mileage of regional highways by the total area of urban districts. A higher number suggests better transportation conditions.

#### 3.2.2. Sample Description

There were 341 observations in the 31 sample cities, with variable information collected from 2002 to 2012. In order to minimize heteroscedasticity, the control variable of economic development, with a large standard deviation, was logarithmically processed. The descriptive statistics of each variable are shown in Table 1.

#### 3.2.3. Data Resources

The API data of the 31 sample cities are based on public data sources, such as the China National Environmental Monitoring Centre [29]. The data on the number of inbound tourists, per capita GDP, number of star-rated hotels, and highway density in 31 cities are from the *China Statistical Yearbook for Regional Economy 2003–2013* [30] and *Statistical Bulletins on the Current National Economic and Social Development* [31]. The number of 4A and 5A attractions was drawn from China’s Ministry of Culture and Tourism website as well as provincial- and city-level cultural and tourism bureaus [32]. As a result, the panel data from multiple sources across 10 years were aggregated and used in the following analyses.

#### 3.2.4. Estimation Formula

As mentioned above, the inbound tourism development degree (INTOURISM) was taken as the outcome variable, and the air pollution index (API) was modeled as the focal explanatory variable. The estimation model was established as follows:INTOURISM*_ij_* = *α_i_* + *β_i_* API*_ij_* + *γ*CV*_ij_* + *ε_ij_*(1)
where *i* is the city; *j* is the period; INTOURISM*_ij_* is the inbound tourism development degree of the *j* period in the *i* region; API*_ij_* is the air pollution level of the *j* period in the *i* region; *a_i_* is the intercept term; *β_i_* and *γ* are the regression coefficients; CV is a group of control variables that may affect the inbound tourism development levels described above; and *ε_ij_* is an independent and identically distributed random error term. This formula was then tested through mixed regression, fixed effects, and random effects.

## 4. Analyses and Results

### 4.1. Data and Variable Test

First, considering the longitudinal nature of the dataset, a test of stationarity was conducted for the explanatory variables. As the *p*-values were all 0.01 or smaller, data stationarity was established. Second, whether there was a long-term stable equilibrium relationship between air pollution and inbound tourism was determined. The results of the Granger causality test indicate that the *p*-value of air pollution on inbound tourism was 0.0517, which is significant at the 10% level. The *p*-value of inbound tourism on air pollution was 0.1962, which is not significant. Thus, the relationship between air pollution and inbound tourism was more likely to stem from air pollution. Thus, the regression analysis that examined the impact of air pollution on inbound tourism was conducted as follows.

### 4.2. The Impact of Air Pollution on Inbound Tourism

In the panel data of 31 cities, the unobservable heterogeneity characteristics of different cities often affected the explanatory variables. The results of the Hausman test further show that the fixed effect model was better than the random effect model for this dataset. Thus, the fixed effect model was selected for regression parameter estimation. In addition, for the purpose of comparative analysis, this paper also reports the random effect (RE) and fixed effect (FE) model results of the regression and discusses the fixed effect model regression results in the analysis process (Table 2). Compared to a bidirectional fixed effect model, the two-way fixed effect model (Model 4) presented a better goodness-of-fit with an adjusted R-squared value of 0.9497. Therefore, the time fixed effect model (Model 4) was used for analysis.

According to the estimation of Model 4, which considers both time and spatial characteristics (as shown in Table 2), air pollution had a significant and negative impact on inbound tourism. For every unit increase in the air pollution index, inbound tourism decreased by 1.171% units. Among the four control variables, the level of tourism resource endowment, tourism reception capacity, and regional economic development showed a significant and positive impact on inbound tourism. Transportation conditions had a significant and negative impact. This might be due to a “crowding out effect” because a higher level of highway density can greatly promote the development of other industries as well as domestic tourism, which may suppress inbound tourism development [33]. Thus, an updated formula (Formula (2)) showing the impact of air pollution on inbound tourism in the YREB is as follows:INTOURISM*_ij_* = *a_i_* − 1.1710 × API*_ij_* + 6.1997 × RESOURCE*_ij_* − 23.3223 × TRANSPORT*_ij_* + 0.2973 × HOTEL*_ij_* + *ε_ij_*(2)

### 4.3. Impact of Air Pollution at Different Times on Inbound Tourism

Based on previous research that considered multiple time periods, the authors of this paper further divided 2002–2012 into three periods, 2002–2005, 2006–2009, and 2010–2012, and the mean values in each period were used for estimation. The results of both the Hausman test and the F test show that the fixed effect model provided the optimal fit. The two-way fixed effect model estimation shows that the *p*-values of the three time periods were 0.179, 0.450, and 0.092, and the regression coefficients of the API were −0.3597, −0.1375, and −0.0291, respectively. That is, the negative impact of the API on inbound tourism was only significant during 2010–2012. This indicates that the impact of air pollution on inbound tourism in the YREB is more prominent in the long term. In addition, according to Figure 2 and Figure 3, air quality has been gradually improving in most cities in the YREB. Thus, it is reasonable to argue that there is a time lag between air pollution and tourism impact as a result of accumulative effects that become more prominent in the long term.

### 4.4. Heterogeneity Analysis

#### 4.4.1. Location Heterogeneity

The authors of this paper further considered the impact of locations. Considering that central cities, such as capital cities, are more likely to attract tourists due to their more recognized destination image and established tourism infrastructure, we divided the sample cities into central cities (municipalities governed directly by the central government, provincial capital cities, and sub-provincial capital cities) and non-central cities (other prefecture-level cities). The results show that when the air quality index increased by 1%, the number of tourists in central cities (*n* = 12) decreased by 1.522%, while the number of tourists in non-central cities (*n* = 19) decreased by 0.5579%. This shows that the impact of air pollution on inbound tourism was more prominent in central cities. This may be because central cities tend to have more media coverage and draw more public attention [34]. Along with the more efficient air quality forecasting systems, tourists are more likely to be informed of air quality issues and make travel decisions accordingly. In contrast, tourists are less likely to be informed of the air pollution levels in non-central cities. Further, central cities are often the gateway cities for tourists to visit other secondary/non-central city destinations. Therefore, it is also likely that tourists will flow to non-central cities when there is a high level of air pollution in the central cities. Lastly, there are more non-central cities than central cities, thus providing more alternatives for tourists when they seek to avoid air pollution.

#### 4.4.2. Tourism Resource Endowment Heterogeneity

As described before, the sample cities were also divided into three groups based on the level of tourism resource endowment, calculated by the weighted number of 4A and 5A attractions in each city. The results show that when the air pollution index increased by 1%, the number of tourists in class A cities (high tourism resource endowment, *n* = 6) decreased by 0.9276%, the number of tourists in class B cities (medium tourism resource endowment, *n* = 8) decreased by 0.8312%, and the number of tourists in class C cities (low tourism resource endowment, *n* = 17) decreased by 0.0641%. This is consistent with the overall positive impact of tourism resource endowment on inbound tourism.

### 4.5. Robustness Test Analysis

We examined the robustness of air pollution to inbound tourism development by eliminating some samples and substituting variables. Yuxi, a low-pollution city, was removed from Model 1, and Chongqing, a high-pollution city, was removed from Model 2. Table 3 shows the regression results of the robustness test. The regression results of the two models show that the significant negative impact of air pollution on inbound tourism development is robust. The coefficient symbols of each variable are similar to those of the previous study. The API has a significant negative impact on the number of inbound tourists at the 1% level. This research shows that air pollution indeed hinders the development of inbound tourism. Therefore, in Model 3, the dependent variable number of inbound tourists was replaced by tourism foreign exchange income (EIT), in which the EIT (learning from inbound tourism) was converted into RMB from the annual average exchange rate of USD to RMB. After replacing the control variables, Model 3 in Table 3 shows that with the increase in air pollution of 1%, the tourism foreign exchange income decreased by 0.5151%, which is significant at the 0.1% confidence level. This again shows that the research conclusion is very robust.

## 5. Discussions

Based on the aggregated panel data of 31 prefecture-level cities in the YREB from 2002 to 2012, this paper articulates the impact of air pollution on inbound tourism. Specifically, we examined the characteristics of air pollution in the YREB, the overall impact of air pollution on inbound tourism in the YREB, and the changes in the impact over time and across regions. In summary, we demonstrate that air pollution has an overall significant and negative impact on inbound tourism in the YREB. For every unit increase in the air pollution index, inbound tourism decreased by 1.171%. However, the impact is not significant in the short term. This indicates a potential lag in the impact of air pollution on inbound tourism. In addition, the impact of air pollution was more prominent in central cities than in non-central cities. Similarly, the impact was stronger for cities with more high-quality tourist attractions.

It is important to acknowledge that we only selected 31 cities as the study sample for the YREB due to data availability. However, these 31 cities include all central cities of the 11 provincial-level administrative regions of the YREB. Considering the strong dispersive nature of air pollution, the air pollution levels in these central cities are likely to be representative of other cities in the region to a significant extent [4]. Future research is encouraged to expand the sample size, as well as time periods, to examine the impact of air pollution.

It is also noted that the results of this study are based on air pollution data before 2013 using the API. While the decision to use the API was largely due to data availability, the extent to which the identified impact of air pollution on inbound tourism in the YREB after 2013 requires more investigation. Considering the potential time lag between air pollution and tourism development, replication studies with AQI data with more recent observations are highly encouraged. Other control factors, such as the Chinese high-speed railway system, which has grown significantly in recent years, should be considered in future research, as it facilitates more long-distance travel. It is also of interest to further consider the types of tourist attractions in different regions. For example, the newly piloted national park system in China may draw more interest from international tourists, and regions rich in world heritage sites are more likely to benefit from inbound tourism. As a result, cities close to these internationally famed touristic attractions are more likely to see the impact of air pollution on inbound tourism. In addition, tourist characteristics and preferences are likely to be significant factors that affect inbound tourist numbers. Tourists who are more adventure-seeking and/or time-constrained are likely to disregard the impact of air pollution. As shown in previous studies (e.g., Zhan and Yin, 2007) [3], air pollution did not show a significant impact on tourism from 2005 to 2012 in China. It is likely that sampling differences may have attributed to this, and such tourist characteristics should be considered in future research.

## 6. Conclusions

This paper contributes to the existing literature by showing how the impact of air pollution on inbound tourism may vary based on time periods, locations, and tourism resource endowment, specifically in the fast-growing YREB. It also adopts a comprehensive air pollution index of the API to account for air pollution levels. The findings offer several valuable implications for mitigating the impact of air pollution on inbound tourism development.

First, it is necessary to have a long-term view of the impact of air pollution on inbound tourism. In the short term, the impact may not be significant. That is, improved air quality does not necessarily result in a significant increase in inbound tourism immediately. Thus, policy-making and marketing forecasting should consider the long-term effect of air pollution or air quality improvement. Second, central cities and cities with more tourism resources should pay more attention to air pollution effects when seeking to promote inbound tourism. Tour operators may consider expanding route options for international tourists to non-central cities and regions with fewer 4A and 5A attractions. When the air quality deteriorates in the main destination cities (e.g., central cities), they have back-up tours ready in other areas. Third, for areas that prioritize inbound tourism development, it is critical to continuously improve air quality. Local economic development and tourism reception capacity also contribute to inbound tourism. In particular, the density of the highway network negatively affects inbound tourism. Thus, promoting alternative routes (e.g., trains, regional flights, river cruises) may help inbound tourism as well.

Lastly, Chinese cities should make full use of tourism resources, improve infrastructure, scale up the development and construction of high-grade tourism resources and service facilities, improve ecotourism services, actively improve the system of ecotourism products and ecotourism industry chain, strengthen the organic integration with other products, and build a supply system of high-quality tourism products and services. It is also necessary to strengthen the publicity and protection of tourism resources, promote civilized tourism, and strengthen the protection and sustainable development of tourism resources.

## Figures and Tables

**Figure 1 ijerph-19-10944-f001:**
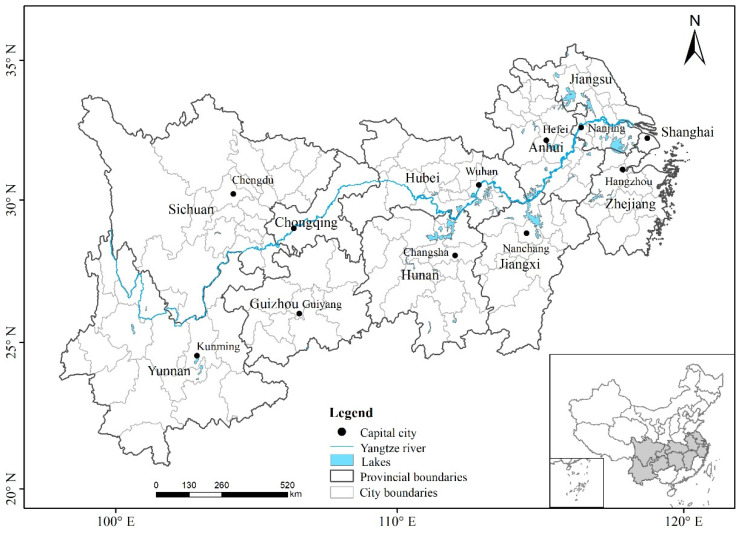
Administrative district map of the YREB.

**Figure 2 ijerph-19-10944-f002:**
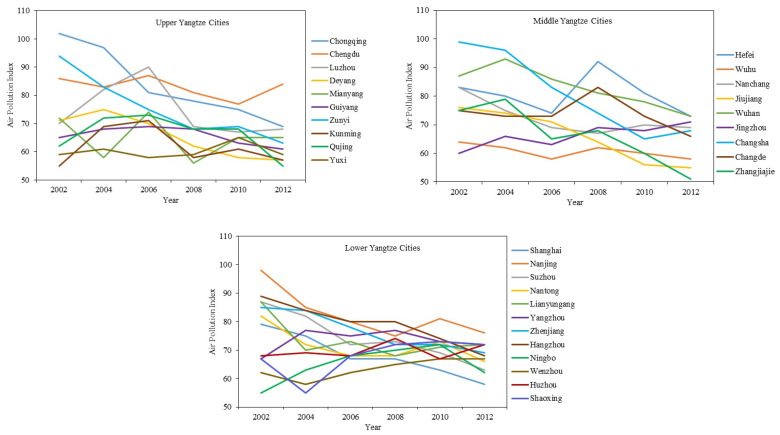
Annual change of air pollution in the YREB from 2002 to 2012.

**Figure 3 ijerph-19-10944-f003:**
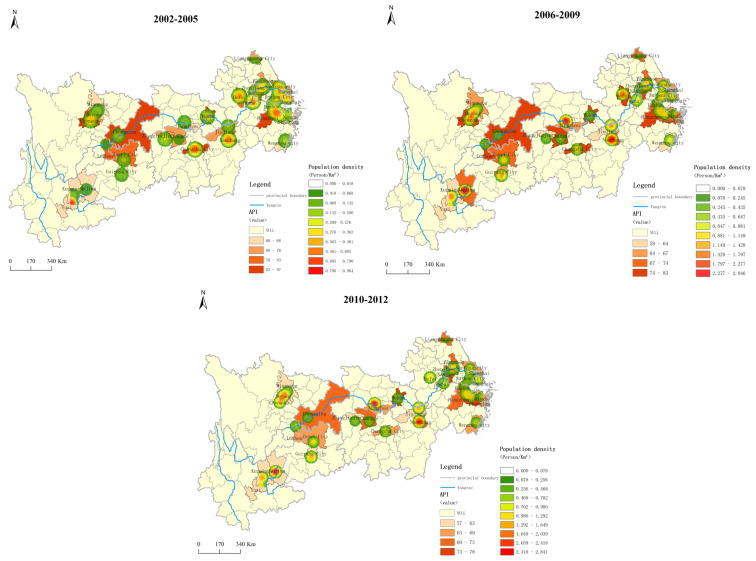
Regional differences of air pollution in the YREB from 2002–2012.

**Table 1 ijerph-19-10944-t001:** Variable descriptive statistics.

Variable Code	Variable	UNIT	Observations	Average	Standard Deviation	Min	Max
NIT *	Number of inbound tourists	Million/ year	341	57.351	118.613	0.05	851.12
API	Air pollution indicators	/	341	70.974	9.239	51	102
RESOURCE	Tourist resources	/	341	6.792	7.185	0	42
TRANSPORT	Highway network density	%	341	1.011	0.469	0.301	2.632
HOTEL	Star hotel	/	341	82.584	70.046	5	359
LNAVGDP	Per capita GDP	RMB/ year	341	10.082	0.754	8.433	11.565
PD	Population density	Person/ km^2^	341	3054.575	2620.99	195	11,562

* NIT is the dependent variable.

**Table 2 ijerph-19-10944-t002:** Regression results of the impact of air pollution and control variables on inbound tourism.

Explanatory Variable	Outcome Variable: Inbound Tourism			
	Pooled OLS (Model 1)	RE (Model 2)	FE (Model 3)	Two-Way Fixed (Model 4)
API	−1.7431 *** (0.5094)	−0.9717 ** (0.3138)	−0.9724 *** (0.2810)	−1.1710 *** (0.2773)
RESOURCE	5.3002 *** (2.0232)	7.8399 *** (0.6247)	7.0898 *** (0.4818)	6.1997 *** (0.5426)
TRANSPORT	−7.4444 (7.3856)	−26.9365 * (12.6764)	−13.3324 *** (5.7607)	−23.3222 *** (14.4938)
HOTEL	1.1821 *** (0.1691)	0.0246 (0.0971)	−0.3568 *** (0.0894)	0.2973 ** (0.0942)
LNAVGDP	73.91194 *** (20.6963)	51.3106 *** (8.9624)	43.8981 *** (6.8412)	41.9745 *** (8.0412)
INTERCEPT	152.9069 * (94.5907)	−26.9365 * (12.6764)	379.1852 *** (61.2400)	876.6027 ** (140.2845)
Adj. R-Squared	0.6994	0.5672	0.9435	0.9497
Location fixed effect				exist
Time fixed effect			exist	exist
Number of samples	341	341	341	341

Note: the values in parentheses are the standard errors, the “***”, “**”, and “*” respectively represent that the coefficient is significant at the levels of 0.1%, 1%, and 5%, respectively.

**Table 3 ijerph-19-10944-t003:** A robust analysis of the impact of air pollution on inbound tourism development from 2002 to 2012.

	Model 1 (Delete Yuxi City)	Model 2 (Delete Chongqing City)	Model 3 (Replace NIT with EIT)
API	−1.4935 ** (0.4598)	−1.5280 *** (0.4529)	−0.5151 *** (0.1433)
RESOURCE	3.9562 *** (1.1349)	3.9540 *** (1.1549)	2.1051 *** (0.2803)
TRANSPORT	−0.0030 *** (0.0006)	−0.0017 (0.0011)	−33.8308 *** (7.4878)
HOTEL	1.2088 *** (0.0924)	1.2403 *** (0.0938)	0.1141 * (0.0487)
AVGDP	0.0001 (0.0003)	0.0000 (0.0003)	19.9451 *** (4.1543)
FDI			359.3664 ** (108.9754)
Adj. R-Squared	0.6708	0.6842	0.7251

Note: the values in parentheses are the standard errors, the “***”, “**”, and “*” respectively represent that the coefficient is significant at the level of 0.1%, 1%, and 5%, respectively.

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
