# Peer review of "Assessing the Impact of Air Pollution on Inbound Tourism along the Yangtze River across Space and Time"

_ijerph, 2022, doi:10.3390/ijerph191710944_

Round 1

Reviewer 1 Report

Overall insightful. Was skeptical at first that the link could be proved but the authors do a good job in the analysis to show the link between tourism and air pollution.

Authors should consider looking at the period after 2013. This will strengthen conclusions.  Or at least indicate why it cannot or isn't necessary.

Reviewer 2 Report

The work correlates the impact of air pollution on tourism along the Yangtze River with the major finding that as pollution intensifies by 1%, inbound tourism decreases by more than 1%.  Another major finding is that the impact to the tourism decrease is delayed in time.

The authors have prepared a very interesting paper which is very well written.  My suggestions for improvement follow.

Lines 140-151:  While the text does state the different conclusions, an additional sentence highlighting these differences would be beneficial to summarize the different conclusions of previous studies.  For example, previous studies range from finding no correlation to PM10 (Zhan and Yin 2017) to PM10 being the second most important factor (Sun 2020).  Perhaps the word "Similarly" in line 149 is what made me read this a second time and suggest a summarization.  Using the word "similarly" I was expecting a similar finding of PM10 not being a factor when indeed it was.  Perhaps another alternative would be to substitute "In contrast," rather than "Similarly".

Line 169:  YREB - In the YREB, there is a higher population density than in China as a whole.  Take one more step in your statistical analysis and consider adding population density to the list of base statistics in this section.

Lines 181-183:  The authors draw a comparison between the YREB emissions and US emissions.  Consider adding a statistic of these emissions globally, not just the US.  Also calculate and compare based on land area and population.  I think this will drive your point even more clearly.

Line 197-198:  It would be helpful to readers to have a paragraph inserted here to explain more about API and how API is calculated.

Line 211 Figure 2:  The graph is hard to read.  The message it sends is that the data is all over the place.  Perhaps consider using a Supporting Information file to add more views of this data and move this chart to Supporting Information.  I think a more meaningful image to share here might be with years on the x axis and broken into 3 graphs with 10 cities (logically organized) in each.  It is just too hard to make comparisons and understand a message from this graph.  

Line 225 Figure 3:  Population density is very important.  Perhaps group this data into three subregions (upper, middle, and lower Yangtze) to better quantify the differences west to east.  These are good map pictures.  Perhaps add the further ArcGIS heatmaps with data grouped as suggested into the Supporting Information file.  Or once you have those new maps, decide if the existing ones should go into the SI file and the new ones into the manuscript.  

Line 273 Table 1:  Note the NIT as the dependent variable.

Line 273 Table 1:  Variables with wide variation, like NIT and HOTEL, typically provide the most impact.  It would be interesting to show impact of HOTEL on NIT.

Line 277-281:  Citations needed for Ministry of Ecology and Environment of China, Statistical Bulletins on the Current National Economic and Social Development, and China's Ministry of Culture and Tourism website.

Line 288 vs 291:  The formula uses alpha sub i, while the text uses a sub i which should likely be alpha sub i.

Line 362-366:  It seems that the regression coefficient indicates that moderately polluted cities show the highest impact on NIT for increases in API.  Perhaps this is because tourists travel to heavily polluted areas despite the pollution since they tend to be gateway cities (called central cities later).  For moderately polluted cities there may be more choice and so the impact shows up better.  Then Section 4.5.1 seems to make the opposite point that the central cities are most sensitive for API impact on NIT.  This needs more thought and consideration to align the message.  Maybe the point is that the tourist sites are more in the non-central cities and the choice is what central city to go through to get to the site.  

Line 366:  The last word may be better as "reduced" rather than "reversed."  As pointed out, the improvement in air pollution has a delayed response, so NIT might actually get worse or stay the same before it gets reduced, and only much later would one expect the improvement in air pollution to result in a reversal.

Line 385: Should "4A and 4A" be "4A and 5A"?

Line 406:  Should this be "at the 0.1% confidence level."?

Line 410:  The title of Model 3 (Replace EIT) is confusing.  I believe you mean "(Use EIT instead of NIT)" or "(Replace NIT with EIT)".

Line 426:  Compare characteristics of the 31 cities to the overall 126 cities for a measure of representativeness back on page 5.  It is still valid that this could be a source of error.

Line 438:  Data might support another alternative.  Maybe cities are not more likely to see the impact of air pollution on inbound tourism because tourists come to famed cities despite API.

Line 456: "contribute" rather than "contributes"

Line 457: "affects" rather than "affected" to maintain present tense as the rest of the paragraph.

Very interesting paper.  It will be an excellent contribution to the literature on this topic.

Round 2

Reviewer 2 Report

The paper, following review, is a very well written and interesting paper.  All suggestions from round 1 were either addressed or rebutted with explanation.